# The Mechanical Microenvironment Regulates Axon Diameters Visualized by Cryo-Electron Tomography

**DOI:** 10.3390/cells11162533

**Published:** 2022-08-15

**Authors:** Di Ma, Binbin Deng, Chao Sun, David W. McComb, Chen Gu

**Affiliations:** 1Ohio State Biochemistry Graduate Program, The Ohio State University, Columbus, OH 43210, USA; 2Department of Biological Chemistry and Pharmacology, The Ohio State University, Columbus, OH 43210, USA; 3Center for Electron Microscopy and Analysis, The Ohio State University, Columbus, OH 43210, USA; 4Molecular, Cellular and Developmental Biology Graduate Program, The Ohio State University, Columbus, OH 43210, USA; 5Department of Materials Science and Engineering, The Ohio State University, Columbus, OH 43210, USA

**Keywords:** axonal varicosity, cryo-electron tomography (Cryo-ET), primary neuron culture, microtubule (MT), mitochondria, multivesicular body (MVB), axon branch point, axon fasciculation

## Abstract

Axonal varicosities or swellings are enlarged structures along axon shafts and profoundly affect action potential propagation and synaptic transmission. These structures, which are defined by morphology, are highly heterogeneous and often investigated concerning their roles in neuropathology, but why they are present in the normal brain remains unknown. Combining confocal microscopy and cryo-electron tomography (Cryo-ET) with in vivo and in vitro systems, we report that non-uniform mechanical interactions with the microenvironment can lead to 10-fold diameter differences within an axon of the central nervous system (CNS). In the brains of adult Thy1-YFP transgenic mice, individual axons in the cortex displayed significantly higher diameter variation than those in the corpus callosum. When being cultured on lacey carbon film-coated electron microscopy (EM) grids, CNS axons formed varicosities exclusively in holes and without microtubule (MT) breakage, and they contained mitochondria, multivesicular bodies (MVBs), and/or vesicles, similar to the axonal varicosities induced by mild fluid puffing. Moreover, enlarged axon branch points often contain MT free ends leading to the minor branch. When the axons were fasciculated by mimicking in vivo axonal bundles, their varicosity levels reduced. Taken together, our results have revealed the extrinsic regulation of the three-dimensional ultrastructures of central axons by the mechanical microenvironment under physiological conditions.

## 1. Introduction

Axon shafts mediate the action potential propagation as well as long-distance transport of various axonal proteins, RNAs, lipids, and organelles, including components of presynaptic apparatuses that are critical for synaptic transmission. The constant diameter along an axon is believed to be crucial to the unidirectional, reliable, and faithful conduction of action potentials [1,2]. Axonal varicosities are enlarged, heterogeneous oblong-shaped structures linked by narrow intervaricosity shafts. They represent a key pathological feature thought to develop via the slow accumulation of axonal damage that occurs during irreversible degeneration, for example in mild traumatic brain injury (mTBI), Alzheimer’s and Parkinson’s diseases, and multiple sclerosis [2,3,4,5,6]. We recently discovered that mechanical stress rapidly and reversibly induced varicosities in the unmyelinated axons of the CNS neurons cultured on glass coverslips, and we further visualized varicosity induction in mouse models of mTBI [7,8]. Our results suggest that microtubule (MT)-associated protein 6 (MAP6, also called stable tubule-only polypeptide) regulates axonal varicosity formation via linking aberrant Ca^2+^ influx through mechanosensitive ion channels to MT depolymerization [7]. Overall, the formation and regulation of axonal varicosities are often investigated in the context of neuropathology [2]. In sharp contrast, the mechanisms underlying axonal varicosity formation in the normal brain remain unknown.

Axonal varicosities are indeed present in the normal brain, although their levels are much lower than those in injured or diseased brains [7,8]. They were found in the unmyelinated axons of the normal brain via three-dimensional (3D) serial electron microscopy (EM) analyses [9,10]. In the hippocampus, most of these axonal varicosities are presynaptic boutons [10]. More recently, using serial block face scanning EM to carry out the 3D reconstruction of axons and myelin, a study showed diameter variation in individual myelinated axons in the optic nerve of rats and in typical white matter [11]. These changes in axon diameter can directly influence the propagation of action potentials. Given the functional significance, it is important to unravel the key factors regulating diameter variation under normal conditions, especially varicosity formation along the axons in the CNS.

Recent studies using various types of super-resolution microscopy have revealed that axons are lined with a periodic scaffold of actin rings that are spaced every 190 nm by spectrins [12,13]. This type of axonal periodic scaffold is present in all of the neuronal types and organisms that have been examined so far as well as in both myelinated and unmyelinated axons [14,15,16]. Altering the activity of axonal non-muscle myosin II can result in changes in the axon diameter of approximately ±30%, indicating the dynamic nature of actin rings [17]. Furthermore, after mechanical unroofing (the removal of the dorsal part of cultured neurons and their axons by ultrasonic pulses), platinum-replica EM (PREM) was used to show the axonal periodic scaffold [18]. However, regular PREM and transmission EM (TEM) failed to reveal or preserve the periodic scaffold of actin rings [13]. Using Cryo-ET, recent studies have also failed to observe the periodic scaffold of actin rings from cultured neurons [19,20]. In Cryo-ET, neurons are vitrified in liquid ethane ultrafast and are analyzed under the microscope at the temperature of liquid nitrogen, eliminating potential artifacts by fixation and organic solvent usage during dehydration and resin embedding. Cryo-ET is becoming a powerful tool to study the ultrastructures of various parts of neurons under normal and abnormal conditions [19,21,22,23,24,25]. Nonetheless, the relationship between the periodic scaffold of actin rings in axons and in axonal varicosities induced by mechanical stress remains unknown.

In the present study, by combining confocal microscopy and Cryo-ET with in vivo and in vitro systems, we found that the formation of axonal varicosities under normal physiological conditions can be regulated by mechanical interactions with the microenvironment. We initially planned to use fluid puffing to induce axonal varicosities in neurons cultured on top of EM grids followed by Cryo-ET analysis, but we unexpectedly found that a substantial amount of axonal varicosities had already formed on the lacey carbon film of the EM grids without fluid puffing. The neurons cultured on the EM grids appeared to be healthy, with normal development. These axonal varicosities likely mimic those observed in the normal brain, whereas the lacey carbon film resembles the non-uniform mechanical microenvironment in the CNS in vivo. Thus, using Cryo-ET in the present study, we focused on the ultrastructure of these axonal varicosities and other enlargements developed on non-uniform surfaces.

## 2. Materials and Methods

### 2.1. Thy1-YFP Transgenic Mice and Brain Tissue Fixation and Sectioning

The Thy1-YFP-H transgenic mouse line was originally purchased from the Jackson Laboratory (Stock # 003782) and has since been maintained in the animal facility at the Ohio State University. The animal protocol (#: 2008A0176-R4) used in the present study was approved by the Ohio State University Institutional Animal Care and Use Committee (IACUC) and conformed to the United States National Institutes of Health Guide for the Care and Use of Laboratory Animals. We have described the procedures of cardiac perfusion, fixation, sectioning, staining, and imaging in detail in our previously published papers [26,27,28,29,30,31]. In brief, three male and three female mice (3–5 months old) were anesthetized with Ethasol (Virbac; 100 mg/kg) and were transcardially perfused with 20 mL of PBS followed by 20 mL of a 4% formaldehyde/PBS solution. Then, mouse brains were removed, post-fixed for 1 h in 4% formaldehyde/PBS solution, cut into 3 mm blocks using an acrylic brain matrix (Braintree Scientific, Braintree, MA, USA), and cryoprotected in a 30% sucrose/PBS solution for 1–3 days. Brain blocks were embedded in optimal cutting temperature (OCT) media (Sakura Finetek USA, Inc., Torrance, CA, USA) and were stored at −80 °C until sectioning. We used a Microm HM550 cryostat (Thermo Scientific, Waltham, MA, USA) to cut brain blocks into 40-μm-thick slides that were then collected on Superfrost Plus microscope slides (FisherScientific, Pittsburgh, PA, USA) for storage at −20 °C, which was maintained using a Microm HM550 cryostat (Thermo Scientific, Waltham, MA, USA). After immunostaining, we mounted the sections with glass coverslips using tris-buffered Fluoro-Gel mounting media (Electron Microscopy Sciences, Hatfield, PA, USA).

### 2.2. Confocal Imaging, 3D Reconstitution and Quantification of Axon Diameter

Fluorescence confocal microscopy and image stack capturing were carried out as described in previous publications from our laboratory [27,29,30,32]. In the present study, we used an Andor Revolution WD spinning disk confocal system (Oxford Instruments, Abingdon-on-Thames, UK) based on a Nikon TiE inverted microscope (60× CFI Plan Apo VC water immersion objective and numerical aperture at 1.40). We captured Z-stack images (8-bit TIFF files) at ~0.10 µm steps for the corpus callosum (Bregma between −1.82 and −2.30 mm) and cortical layer VI (Bregma between +1.98 and +1.78 mm). Image stacks were analyzed with NIH ImageJ. Axon diameters were measured on the Z-projection after Dynamic Reslice. The diameter values from the thickest and thinnest portions in the same axonal segment were compared and plotted.

### 2.3. Hippocampal Neuron Culture on Cover Glasses, Immunostaining and Imaging

We prepared the primary culture of mouse hippocampal neurons from mouse pups at postnatal days 0–1 (P0–P1) using the same procedure as previously described [33,34]. We used rat tail collagen and poly-D-Lysine to coat glass coverslips. In brief, at 2 days in vitro (DIV), we added 1 μM of cytosine arabinose (Sigma-Aldrich, St. Louis, MO, USA) to neuronal culture media to inhibit glial growth and then replaced the media with normal neuronal culture media two days later, at 4 DIV. For transient transfection, we incubated cultured neurons at 5–7 DIV in Opti-MEM containing 0.8 μg of cDNA plasmid and 1.5 μL of Lipofectamine2000 (Invitrogen, Carlsbad, CA, USA) for 20–30 min at 37 °C.

### 2.4. Hippocampal and Cortical Neuron Culture on EM Grids and Vitrification

Gold EM grids (200 mesh; with ~97 μm grid hole size) with lacey carbon film (Cat. #: 01882G; Ted Pella INC, Redding, CA, USA) were coated in the same way as the coverslips described in the prior section. The size of most of the holes in the lacey carbon film was between 0.25 and 10 μm. Hippocampal or cortical mouse neurons were cultured on these gold-coated EM grids. At 10–12 DIV, neurons growing on the grids were washed with PBS. Extra liquid was carefully blotted away from the sides of the EM grids where neurons were not adhered using a piece of filter paper. The grids were immediately flash frozen using a homemade manual Cryo-plunger. The frozen grids were transferred and stored at the temperature of liquid nitrogen in a storage tank.

### 2.5. Cryo-Electron Tomography (Cryo-ET)

The frozen grids were screened using Thermo Scientific™ Glacios™ Cryo-TEM. Grids with suitable ice thickness and an adequate density of neurons were transferred to a Thermo Scientific™ Titan Krios™ for Cryo-ET data collection. The Titan Krios was equipped with an AMETEK Gatan’s K3™ direct detector and AMETEK Gatan™ energy filter. Cryo-ET data were collected at 300 kV, with a pixel size of 0.39 nm/pixel at a magnification of 19,500×. The tilting range was from −60 degrees to 60 degrees. Data were collected at 3- or 4-degree intervals. A dose-symmetric tilt scheme was applied as previously described [35]. The total dose for one data set was 80–100 electrons per square angstrom.

### 2.6. 3D Reconstruction and Segmentation

The tomography data were processed using IMOD software as previously described [36]. The reconstructed 3D structures were segmented and visualized using Thermo Scientific™ Avizo software. Cell membranes, mitochondria, endoplasmic reticulum, and irregular vesicles were manually segmented by selecting areas containing corresponding structures using segmentation tools in Thermo Scientific™ Avizo software to generate a smooth surface view after segmentation. Microtubules and round vesicles were manually segmented using the filament function in Thermo Scientific™ Avizo software.

### 2.7. Puffing-Induced Axonal Varicosity Formation and Live-Cell Imaging

We used our fluid puffing system and live-cell imaging as described in our previous papers [7,37,38]. To provide local fluid puffing, we connected the glass pipette (tip diameter ~50 μm) to a syringe via tubing filled with 20 mL of Hank’s buffer and elevated the syringe 190 mm above the tip of the pipette. We positioned the pipette tip 0.4 mm (vertical distance) above the cultured neurons. We defined an axonal varicosity as when a diameter was ≥300% than the diameter of its adjacent axonal shafts. The varicosity onset time was defined as the time for an axonal segment to reach 10 varicosities per 100 μm of length during puffing. Of note, under normal conditions without puffing, the axon diameters were not perfectly uniform in the presence of a low number of varicosities. Thus, the baseline of varicosity density along the axons in culture is not absolute zero.

## 3. Results

### 3.1. Diameter Variation of Individual Axons Correlates with the Complexity of Microenvironment

To determine diameter variation or varicosity abundance in axons in the white and gray matter of the brain under normal conditions, we performed confocal microscopy in the corpus callosum and somatosensory cortex of adult Thy1-YFP transgenic mice. Confocal image stacks captured with a 60× lens were reconstituted to be 3D in order to accurately measure the diameters along individual YFP-positive (YFP+) axons. Bundles of myelinated axons give rise to a relatively light appearance of some brain regions, the so-called white matter, including in the corpus callosum. Our results showed significant diameter variation (D_max_/D_min_: 1.0–3.0; *n* = 100) along individual YFP+ axons (Figure 1A). Along a single continuous segment of a YFP+ axon, we measured the largest diameter (D_max_) and smallest diameter (D_min_) to obtain the ratio of D_max_/D_min_ to reflect diameter variation. Thus, even in the highly compacted axonal bundles in the corpus callosum, typical white matter in the brain, the diameters along individual axons are not perfectly uniform. Next, we examined YFP+ axons in the cortex, typical gray matter in the brain. The range of diameter variation along individual YFP+ axons in the cortex (D_max_/D_min_: 1.3–13.1; *n* = 99) was remarkably larger than in the corpus callosum (Figure 1B). Among approximately 200 axonal segments from the two brain regions, the D_max_ and D_min_ values correlated well in the corpus callosum (R^2^ = 0.69), whereas such correlation was poor in the cortex (R^2^ = 0.02) (Figure 1A,B lower panels). In the cortex, there were many more thin axons with varicosities, and most of these varicosities presumably represented presynaptic boutons (Figure 1A,B), as previously revealed in the hippocampus [10]. Taken together, our results suggest that gray matter axons may display more diameter variation than white matter ones. In the normal brain, the microenvironment is much more complicated in gray matter than white matter. For instance, gray matter contains neuronal cell bodies, dendrites, and axons as well as a host of other cell types. In contrast, white matter contains axons but not neuronal cell bodies and dendrites. Thus, our results raised previously unknown questions as to whether and how microenvironment complexity regulates diameter variation in individual axons.

To determine potential intrinsic and extrinsic factors that regulate diameter variation in individual CNS axons, we used primary cultures of hippocampal and cortical mouse neurons combined with Cryo-ET. Rodent primary neuron culture represents the most popular model for cell biological studies of neurons [39,40,41,42]. Hippocampal or cortical neurons from mouse pups (0–1 postnatal days) were differentiated on coated glass coverslips or on the plastic bottom of a cell culture dish as described previously [7,28,33]. At 1 DIV, the neurons demonstrated a few processes; at 3 DIV, one of the processes became significantly longer than the other axons; at 7 DIV, both the axons and dendrites were further elongated; at 14 DIV, the axons and dendrites formed complicated networks, and the synapses started to form (Figure 1C). To visualize the 3D ultrastructure of the axons from these cultured neurons, we cultured them on gold-coated EM grids (Figure 1D). These grids had a thin lacey carbon film layer with holes (0.25–10 μm diameter) to support the disassociated neurons (Figure 1E). On top of the lacey carbon film that was coated with extracellular matrix protein collagen and poly-D-lysine, neurons were still able to differentiate and grow long processes, both axons and dendrites, similar to on the coverslips (Figure 1F). Low-magnification Cryo-EM images showed that whereas electrons cannot pass through thick cellular structures, including neuronal soma and dendrites, there were still many areas with thin and isolated axons that could be resolved in Cryo-ET (Figure 1G). Therefore, our analysis focused on isolated axons with diameters ≤2 μm.

To determine whether the ultrastructures of the axons growing on the flat surface or on the lacey carbon film were similar, we performed TEM imaging of the neurons cultured on coated plastic bottoms of 24-well plates and Cryo-EM imaging of the neurons cultured on coated EM grids with a lacey carbon film. Our TEM images showed that although not perfectly uniform, the axons growing on the flat plastic surface had rather consistent diameters (Figure 1H left). In sharp contrast, the Cryo-EM images showed axons growing on the lacey carbon film developed numerous varicosities that were exclusively located within the holes (Figure 1H right). Therefore, even without external fluid mechanical stress, axonal varicosities were effectively induced on the non-uniform supporting surface.

### 3.2. Ultrastructures of Axonal Varicosities Induced by Non-Uniform Extracellular Support

To visualize the ultrastructures of axonal varicosities from the CNS neurons cultured on coated lacey carbon films, we performed Cryo-ET on approximately 40 axonal varicosities of various sizes. All of these varicosities were located in the holes of the lacey carbon film and contained at least 1 MT filament. We categorized them into several groups based on their contents. The first type of axonal varicosities contained mitochondria as well as a number of tiny vesicles (diameters ~ 50 nm) and eight MTs with different bending features (Figure 2A–C and Appendix A). This type of axonal varicosity with mitochondria appeared to be the most abundant one. The second type was varicosities containing one large multivesicular body (MVB), a single membrane enveloping many tiny vesicles (Figure 2D,E). This particular varicosity also contained five MTs and many tiny vesicles (diameter 50–100 nm) (Figure 2D,E and Appendix A).

To quantify the effect of the lacey carbon film on diameter variation in cultured axons, we used Cryo-EM images and measured the largest diameter of a varicosity (D_varico_) in a hole of a carbon film and the smallest diameter along the nearby axonal shaft (D_shaft_) on top of the carbon fiber. Among the over 100 axonal segments that we examined, the D_varico_-D_shaft_ correlation was fair (R^2^ = 0.22; *n* = 191), and the D_varico_/D_shaft_ ratios were between 1.7 and 17.4 (Figure 2F), similar to the large range of D_max_/D_min_ in the cortex (Figure 1B). Thus, lacey carbon film can be used to mimic the non-uniform microenvironment in gray matter in the brain in terms of promoting diameter variation in individual axons.

To determine whether any MT disruption occurred in the axonal varicosities on the lacey carbon film, we carefully examined the MTs in both the varicosities and their nearby intervaricosity shafts. We did not observe even a single varicosity with increased or decreased MT numbers compared to the nearby shafts. All of the varicosities contained the exact same number of MTs as their nearby shafts. We did not observe a single MT free end from these axonal varicosities (Figure 2A–E). This is markedly different from the axonal varicosities, which showed MT disruption induced by fluid mechanical stress, injuries, or diseases [2,7]. Thus, these axonal varicosities that formed on the lacey carbon film mimic those formed in the normal brain and may be reversible. Interestingly, the numbers of MTs in axonal shafts actually correlated with D_shaft_ nicely (R^2^ = 0.81; *n* = 76) (Figure 2G). When their diameters were smaller than 50 nm, most of these axon shafts that were most likely localized in distal axons did not contain a single MT (Figure 2G).

We further noticed that compared to their nearby shafts, the axonal varicosities often contained more subcellular organelles despite having the same number of MTs. Approximately half (49%) of the axonal varicosities contained at least one mitochondrion, whereas 16% of them contained MVBs, and 5% contained both mitochondria and an MVB (Figure 2H). Approximately 40% of the axonal varicosities did not contain either mitochondria or an MVB (Figure 2H), but most of them still contained many tiny vesicles. This result is consistent with the results of an early study using 3D serial EM analysis [9].

The axon shafts on top of the carbon fiber were often the narrowest portion of the axon, without much extra space besides the MTs inside of it, hence causing it to pose a tremendous challenge for the axonal transport of large organelles. In particular, since the mitochondria were enriched in axonal varicosities, it appeared to be quite difficult for them to be transported to other parts of the axon. Interestingly, we observed that a mitochondrion might be at the transition of passing through a narrow axonal shaft on top of the carbon fiber with its stretched outer membrane and four MTs through the shaft (Figure 3 and Appendix A). It is also possible that the mitochondrion was undergoing a fission process.

### 3.3. A Majority of Puffing-Induced Axonal Varicosities Formed around Mitochondrial Clusters

When neurons are cultured on a flat surface, their axons usually display smooth and relatively uniform diameters. Our recent studies showed that fluid puffing can rapidly reverse varicosity formation on these axons [7]. Here, we used the same system with the minimal puffing pressure (190 mm H_2_O) that is still able to reliably induce axonal varicosities, but we reduced the puffing time to 50 s. The shorter puffing time appeared to induce axonal varicosities with smaller sizes and better recovery.

The axonal transport of mitochondria has been extensively studied before [43,44]. To determine the spatial relationship between puffing-induced axonal varicosities and mitochondria, we transfected neurons with mito-YFP that specifically labeled the mitochondria as YFP+ clusters along axons (Figure 4A). During puffing, axonal varicosities started to form at the sites with preexisting mitochondria, as indicated by the YFP+ clusters; when puffing stopped, most of the YFP+ clusters remained at the same location (Figure 4A,B). In the kymograph, there was clear retrograde movement before and after puffing, with YFP+ puncta indicating that the varicosities that formed during puffing were located at the preexisting sites where mito-YFP had already clustered (Figure 4B). In contrast, the EB1-YFP proteins were concentrated into axonal varicosities at new sites during puffing and were redistributed throughout the axons afterwards (Figure 4C left). EB1-YFP is used to reveal plus-end tracking in MTs, but under the over-expressed condition, there is a large pool of EB1-YFP molecules not binding to MTs. These free-floating molecules can be used to indicate the volume of the axon, just like YFP molecules. There was no apparent preference for MT plus-end tracking into or out of the varicosities (Figure 4C left), suggesting that MT plus-end tracking unlikely plays a key role in varicosity formation. Transiently expressed APP-YFP displayed fast anterograde axonal transport both before and after puffing; during puffing, the percentage of axonal varicosities formed at the sites with preexisting YFP+ clusters was higher than those for YFP and EB1-YFP; when puffing stopped, some YFP+ clusters remained at the same location for a long time (Figure 4C right). Thus, most of the puffing-induced axonal varicosities formed at the sites already containing preexisting mito-YFP, but this was not the case for YFP or EB1-YFP (Figure 4D).

To simultaneously visualize axonal varicosity formation and mito-YFP punctum movement, we performed live-cell imaging with transmitted light (black in gray scale) and YFP fluorescence (green) (Figure 4E). Our result showed that most axonal varicosities (swellings revealed under transmitted light) formed at the sites with preexisting mitochondrial clusters (mito-YFP puncta revealed by YFP fluorescence), whereas only a small fraction of induced axonal varicosities formed without containing any YFP fluorescence signal (Figure 4E,G). Furthermore, we examined YFP-Rab7, which specifically labels late endosomes and MVBs [45,46]. When being expressed in neurons, YFP-Rab7 fluorescence signals were distributed more smoothly throughout axons with sparse clusters where MVBs were most likely localized (Figure 4F). After puffing, some axonal varicosities formed at sites where YFP-Rab7 was clustered, while most varicosities formed at the sites that did not contain YFP-Rab7 clusters before puffing (Figure 4F,G). Thus, comparing mito-YFP and YFP-Rab7, our results indicate that most of the axonal varicosities formed at the sites where mitochondria were located, consistent with our finding that mitochondria were enriched in axonal varicosities when the neurons were cultured on top of the lacey carbon film (Figure 2).

### 3.4. Axon Branch Points Often Possess Enlarged Diameters and Are Highly Heterogeneous

To search for diameter variations along individual axons, we found that axon branch points were another type of axonal site with an enlarged diameter and exclusively found in the holes of lacey carbon film. Axon branching plays a key role in proper neural network formation [47]. Axonal branches are proposed to form in two major ways: (1) splitting of a growth cone and (2) de novo branching from the axon shaft, called collateral branch formation, which appears to be the major mechanism [48]. In the present study, we performed Cryo-ET at approximately 20 axon branch points. A common feature for all of these branch points was the unequal split of MT filaments into daughter branches, suggesting that they belong to collateral branch formation. In one typical type of branch points, all four preexisting MTs extended into what presumably was the main branch (the main daughter axon), while two additional MTs were formed de novo in the other branch (the thinner daughter axon), with their free ends located within the branch point (Figure 5A–C and Appendix A). The free ends of the two MTs were presumably the minus ends. This branch point also contained a large mitochondrion (17 out of total 35 branch points in the present study contained at least one mitochondrion) and tiny vesicles (Figure 5B). In another typical type of axon branch point, all three MTs extended into the main branch, whereas the other thin daughter branch contained no MT near the branch point but did contain MTs in the region further away from the branch point (Figure 5D–F and Appendix A). The diameter of this thin branch was quite small, close to the diameter of a MT filament (30–40 nm). This branch point contained a number of vesicles of different sizes (Figure 5E,F). Similar to the axonal varicosities that we visualized with Cryo-ET, these branch points often contained mitochondria, vesicles, and MVBs. Different from those varicosities, we did frequently observe MT free ends within axon branch points. These MT free ends led to MTs in the daughter branch. If the two types of axon branch points (Figure 5) represent two different developmental stages, then our Cryo-ET results further suggest that the thin and collateral axon branch may not contain an MT initially, and MT(s) may be added after the branch is stabilized.

When extending out of the main axon, the thin branch sometimes also contained MT filaments. In an example, out of a total of the six MTs in the main branch, five MTs remained in the main branch, whereas one MT extended into the thin branch (Figure 6A,B). The MT extending into the thin branch displayed a high degree of bending (Figure 6B). This is consistent with the notion that MTs can bend significantly when being modulated [49]. To determine the distribution of the tubulins in the axon branch points from the neurons cultured on flat uniform surface, we stained them with an anti-β-tubulin antibody (green) and with phalloidin (red) for the actin filaments. The β-tubulin staining signal profiles along all of the axonal branches often showed reduced overall intensity in a minor branch (Figure 6C), consistent with a reduced number of MTs in the thin branch. We also frequently observed that the beginning portion of the minor branch often had lower staining signals compared to its more distal portion (Figure 6D), consistent with what we observed in Cryo-ET (Figure 5D–F). Furthermore, we observed clusters of phalloidin staining signals along axons (Figure 6C,D). These clusters were likely actin hotspots that were present throughout the axon shafts [50].

### 3.5. Bundled Axons Tend to Have Relatively More Uniform Diameters with Fewer Axonal Varicosities

Axons often fasciculate to form bundles in the brain, giving rise to white matter. When neurons are cultured in vitro over a flat and uniform surface, their axons frequently form bundles with occasionally separated segments (Figure 7A). When cultured on the lacey carbon film of the EM grids, the axons could also form bundles (Figure 7B). These axons still formed varicosities in the holes, but the frequency was significantly reduced, especially when multiple axons fasciculated in the hole (Figure 7B,C). Ultrastructural Cryo-ET images showed that the plasma membranes of the two fasciculated axons were close to each other and that they had MTs running in parallel (Figure 7D–F and Appendix A). The unrestricted side of an axon could still form a varicosity in the hole, and the axon–axon contact was not uniform (Figure 7B–D). Thus, in vitro axon fasciculation appeared to reduce the abundance of axonal varicosities, consistent with the lower level of axonal varicosity formation in white matter compared to gray matter in vivo.

## 4. Discussion

In the present study, using Cryo-ET with hippocampal/cortical neurons cultured on lacey carbon film on gold EM grids, we show that non-uniform external support alone can lead to axonal varicosity formation with diameters that are up to 13–17 times larger than those in nearby shafts and that axon fasciculation reduces such formation (Figure 1, Figure 2, Figure 3 and Figure 7). These results partially explain why gray matter axons possess higher variability in diameter than white matter ones in the normal brain. Taken together, our Cryo-ET results have revealed for the first time that the 3D ultrastructure of axonal varicosities is induced by their mechanical microenvironment.

Our high-resolution confocal imaging stacks allowed us to measure diameters along individual axons and revealed significant diameter variation in individual axons in the cortex and, to a lesser extent, in the corpus callosum (Figure 1A,B). On the other hand, most early studies on axon diameter used the 2D analysis of axon cross sections within a brain region, and the same axon could be assigned into multiple different groups based on the value of the diameter during analysis. Importantly, our results suggest that gray matter axons are more likely to have varicosities or uneven diameters than white matter ones. In the brain, gray matter but not white matter contains neuronal cell bodies, dendrites, and synapses. White matter contains orderly bundled myelinated axons as well as unmyelinated ones, giving rise to its white appearance. Gray matter also contains axons, with a small percentage of them even being myelinated, but these axons have mixed orientations, forming mesh-like patterns. In addition to their differences in terms of neurons, gray matter and white matter differ in their non-neuronal cells, such as astrocytes [29]. Overall, the microenvironment of axons is much more complex in gray matter than white matter in terms of mechanical and chemical properties. Our in vitro data suggest that the mechanical microenvironment alone can lead to diameter variations in individual axons. Although they were coated with the exact same materials to rule out a direct role of chemical cues, the lacey carbon film but not the flat surface induced large axonal varicosities (Figure 1C–H). Furthermore, when being cultured on a smooth and flat surface, the axons developed varicosities in response to fluid puffing (Figure 4) [7]. Finally, reduced levels of axonal varicosities in fasciculated axons on top of lacey carbon film also support a critical role of the mechanical microenvironment in varicosity formation (Figure 7).

Cryo-ET can become a powerful ultrastructural tool for cell biological studies of neurons [19,21,22,23,24,25]. Different from room temperature TEM, in Cryo-ET, neurons are vitrified ultrafast in liquid ethane and are analyzed under the microscope at the temperature of liquid nitrogen, which eliminates potential artifacts generated by chemical fixation and organic solvent usage during dehydration and resin embedding, such as the expansion and shrinkage of cellular materials. Moreover, the average Cryo-ET subtomogram has the potential to solve the protein structures within intact cellular contexts. Nonetheless, despite its unique strengths, Cryo-ET is unlikely to replace other imaging techniques. For instance, we have been wondering what the relationship between periodic actin rings and axonal varicosities is. In axons, actin rings are evenly spaced every 190 nm by spectrins [12,13], whereas the distances between two adjacent axonal varicosities induced by mechanical stress are highly variable and approximately 4000 nm on average [7,8]. It would be interesting to determine whether varicosities and narrow shafts possess actin rings of different sizes, which may, in part, provide membrane tension. The actomyosin network underneath the plasma membrane and neuronal activity were proposed to control and regulate axon diameter, but diameter changes were less than one-fold or 100% in these studies [17,51,52]. In sharp contrast, the diameter variations were up to approximately 13-fold in the cortex in vivo and 17-fold in the neurons cultured on the lacey carbon film in the present study (Figure 1 and Figure 2). Thus, these results from different groups are likely the result of focusing on different types of axonal varicosities. Furthermore, our Cryo-ET studies do not reveal clear ring-like ultrastructures for actin filaments, consistent with two other recent studies using Cryo-ET [19,20]. Therefore, this question remains to be addressed in future studies, perhaps by using a combination of advanced imaging techniques.

In the present study, we report a novel strategy (lacey carbon film) to induce numerous axonal varicosities in cultured CNS neurons. Most cell biological studies of CNS neurons have used primary neuron cultures on flat glass or plastic surfaces, which does not mimic any structure in any region of the brain. In contrast, lacey carbon film provides a simple binary microenvironment that includes a liquid/solid interface on the carbon fiber and a liquid-only environment supported by culture media in the hole. This partially mimics the mechanical microenvironment in vivo since different CNS cell types have different mechanical properties in terms of stiffness. Thus, our results suggest that varicosities preferentially form at axonal sites with the softest microenvironment. It is important to note that our CNS neurons were cultured on lacey carbon film under normal conditions, and these neurons were at least morphologically indistinguishable from the ones cultured on glass or plastic surfaces (Figure 1). The physical interactions between the carbon fibers and axons were mainly mediated by the electrostatic forces between the coated poly-D-lysine and the cell membranes, whereas the gravity might play a rather minor role since the cells were surrounded by culture medium. The non-uniform external support from the lacey carbon film is actually more consistent with the in vivo situation, and mimics the cavities and microcavities observed during CNS development and injury in particular [53,54]. Thus, axonal varicosities may have the potential to be developed as a novel indicator to estimate the microcavity or the softness in various brain regions.

Our results show that mitochondria were primarily present in the axonal varicosities in the holes of lacey carbon film, consistent with their presence in puffing-induced axonal varicosities (Figure 2, Figure 3 and Figure 4). Another type of large organelle, MVBs, was also present in the axonal varicosities, but to a lesser extent than mitochondria. On the other hand, small vesicles were observed in both the varicosities and shafts, and their enrichment in the varicosities likely resulted from the larger total volume there. It is important to note that narrow axon shafts pose a tremendous challenge for MT-dependent axonal transport, especially for large organelles such as mitochondria and MVBs. This is because our Cryo-ET results show that the diameters of the axon shafts were often not much larger than the combined diameters of the MTs inside, leaving very limited space for anything else (Figure 2 and Figure 3). Axonal transport will be extremely difficult without dynamic changes in axon diameters, consistent with the notions of recent studies [51]. Thus, axonal transport is likely driven not only by molecular motors, but also by external forces from the microenvironment to dynamically shape axon diameters.

Using Cryo-ET, we found that all of the axonal varicosities on the lacey carbon film contained intact MTs or the same number of MTs as the nearby shafts, and there was no sign of MT disassembly (Figure 2 and Figure 3). This result is different from our early findings showing disassembled MTs in some axonal varicosities induced by fluid puffing [7]. Axonal varicosities are highly heterogeneous. Our results show that the range of the D_varico_/D_shaft_ ratios of the axonal varicosities that formed on the lacey carbon film was between 1.7 and 17.4 (Figure 2). This may still fall into the normal or physiological range (Figure 1A,B). On the other hand, the axonal varicosities induced by fluid puffing may possess a much wider range of diameter variation in the D_varico_/D_shaft_ ratio (e.g., >20). Interestingly, a recent computational study suggested that the relative sizes of axonal varicosities are important for altering action potential propagation, and propagation failures start when the ratio is higher than approximately 20 [55]. If so, those axonal varicosities that we observed in the cortex and corpus callosum and over the lacey carbon film (Figure 1 and Figure 2) are unlikely to prohibit action potential propagation. On the other hand, some larger axonal varicosities with disassembled MTs may cause action potential propagation to fail and represent a pathological condition for the axons.

Axon branch points were another type of enlarged sites along the axon shafts that formed exclusively in the holes of the lacey carbon film. They were similar to axonal varicosities in many aspects, with the exception that we only found MT free ends within some of the axon branch points (Figure 5 and Figure 6). These MT free ends led to MTs inside a thin daughter branch. As a result, these daughter branches did not demonstrate significant MT bending. It remains unclear whether varicosity formation precedes branching at the axon branch point or if the enlargement at a branch point only takes place after branching. Moreover, in all of the axon branch points that we examined, we never found an example of MT branching.

## 5. Conclusions

In summary, our findings from the present study have provided multiple novel insights into the axonal properties of central neurons in response to their mechanical microenvironment under normal conditions. At the same time, our results also raised some interesting questions regarding the regulation of the 3D ultrastructures of axons for future investigation.

## Figures and Tables

**Figure 1 cells-11-02533-f001:**
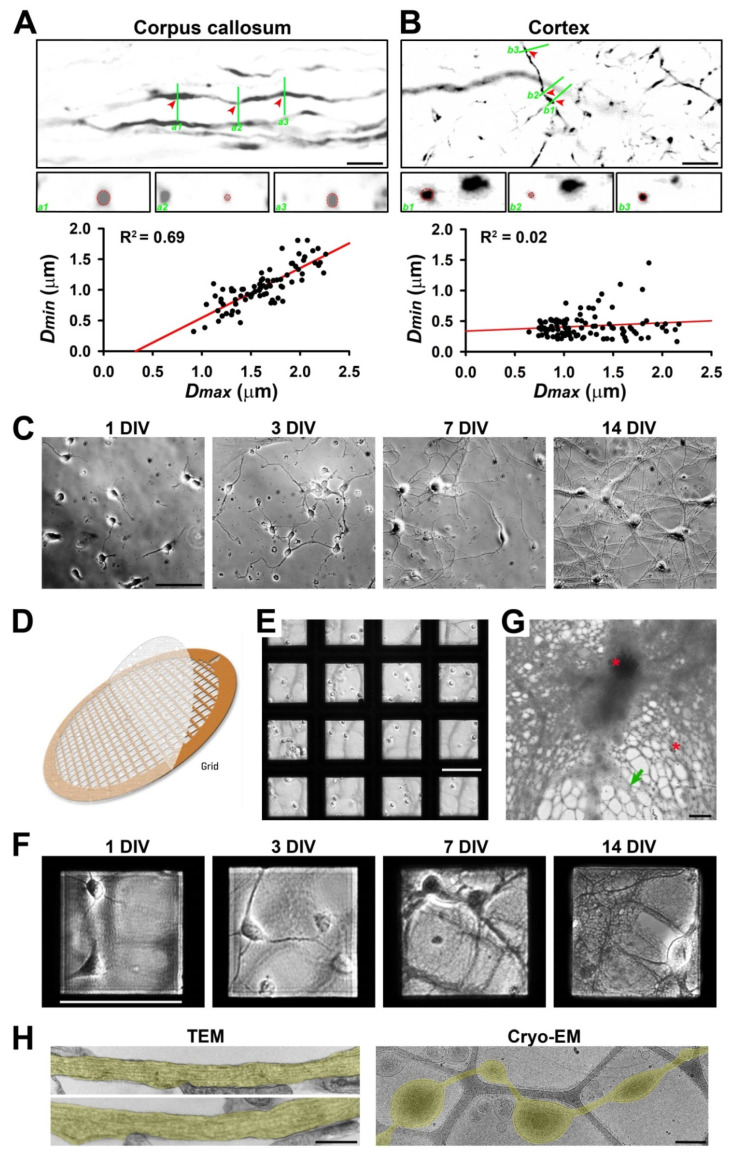
**Diameter variation in individual axons correlates with the complexity of extracellular microenvironment.** (**A**) Confocal images and 3D reconstruction of YFP+ (signals inverted in gray scale) axons in the corpus callosum of Thy1-YFP transgenic mice. In an XY focal plane (top), three green lines indicate the positions for three ZY focal planes (a1–a3, middle, enlarged by 2-fold) to show the diameters of three positions (indicated by red arrowheads) of a single axonal segment (circled in red dashes). Correlation between the largest diameter (D_max_) and the smallest diameter (D_min_) along a single continuous axonal segment is indicated by a black dot in the chart at the bottom. (**B**) Confocal images and 3D reconstruction of YFP+ axons in the cortical layer 6 of Thy1-YFP transgenic mice. The D_min_-D_max_ correlation in the cortex is much less than that in the corpus callosum. (**C**) In vitro development of mouse hippocampal neurons cultured on coated glass coverslips. Neurons at 1, 3, 7, and 14 days in vitro (DIV) were visualized with transmitted light. (**D**) The diagram of an EM grid with lacey carbon film. (**E**) Neurons were just seeded on coated EM grids with lacey carbon film at 0 DIV. (**F**) In vitro development of hippocampal neurons cultured on EM grids. Neurons at 1, 3, 7, and 14 DIV were visualized with transmitted light. (**G**) Low-magnification Cryo-EM image of neuronal processes growing on coated lacey carbon film. Red asterisks, ice crystals. Green arrows, thin and isolated axons that can potentially be resolved in Cryo-ET. Blurry areas represent thick cellular structures. (**H**) TEM images of axons growing on a coated and uniform plastic surface (**left**) and Cryo-EM image of an axon growing on lacey carbon film (**right**). Axonal segments are highlighted in yellow. Scale bars: 10 μm in (**A**,**B**,**G**), 100 μm in (**C**,**E**,**F**), and 500 nm in (**H**).

**Figure 2 cells-11-02533-f002:**
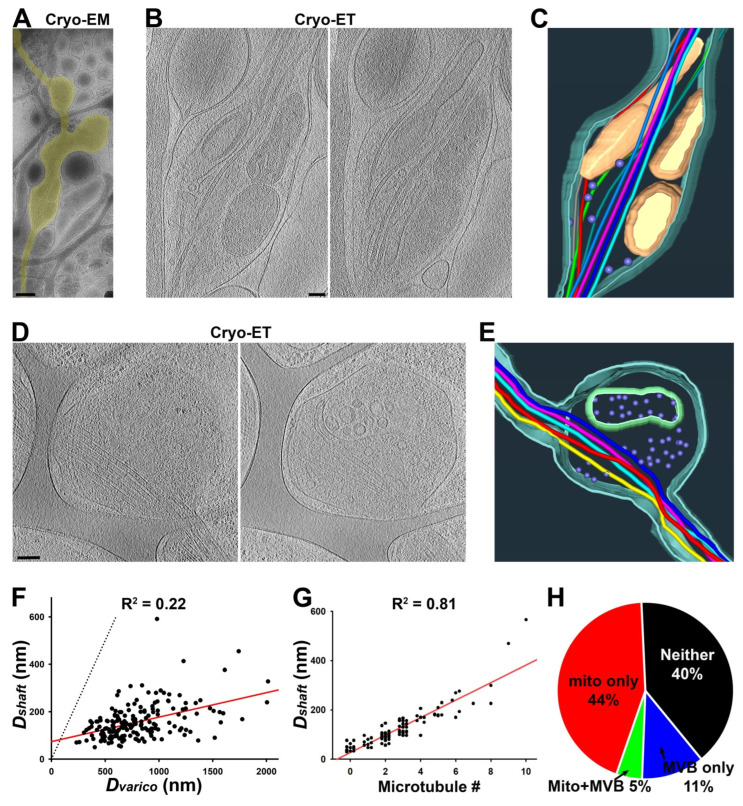
**Three-dimensional ultrastructures of axonal varicosities on lacey carbon film revealed by Cryo-ET.** (**A**) Low-magnification Cryo-EM image of an axon (highlighted in yellow) growing on lacey carbon film. (**B**) Two slices from the Cryo-ET tomogram of the axonal varicosity. (**C**) Segmentation view of the varicosity in (**B**). MTs were represented in tubes of different colors: mitochondria in yellow, plasma membrane in cyan, and small vesicles in purple. (**D**) Two slices from the Cryo-ET tomogram of an axonal varicosity containing an MVB. (**E**) Segmentation view of the varicosity in (**D**). The outer membrane of the MVB is in green. (**F**) The correlation between the diameter of the axonal varicosity (D_varico_) and the diameter of its nearby shaft growing on top of carbon fiber (D_shaft_). (**G**) The correlation between the diameter of the axonal shaft (D_shaft_) and the number of its MTs. (**H**) Percentages of axonal varicosities containing mitochondria but not an MVB (red, 44%), MVB(s) but no mitochondria (blue, 11%), both mitochondria and an MVB (green, 5%), and neither mitochondria nor an MVB (black, 40%). *n* = 191. Scale bars, 500 nm in (**A**) and 100 nm in (**B**,**D**).

**Figure 3 cells-11-02533-f003:**
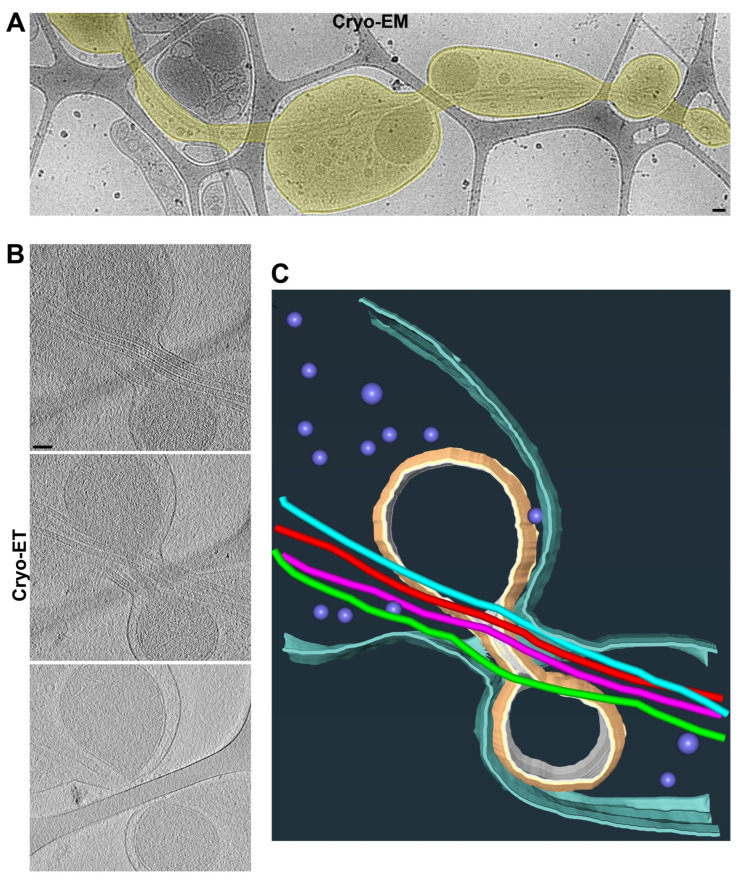
**Two adjacent axonal varicosities with carbon fiber support in the middle.** (**A**) Low-magnification view of an axonal segment with multiple varicosities growing on top of lacey carbon film. The axon is highlighted in yellow. (**B**) Three slices from the Cryo-ET tomogram of the junction of two axonal varicosities under high magnification. (**C**) Segmentation view of the varicosity junction in (**B**). Four MTs are represented as tubes of four different colors: mitochondria in yellow, plasma membrane in cyan, and small vesicles in purple. Scale bars, 100 nm.

**Figure 4 cells-11-02533-f004:**
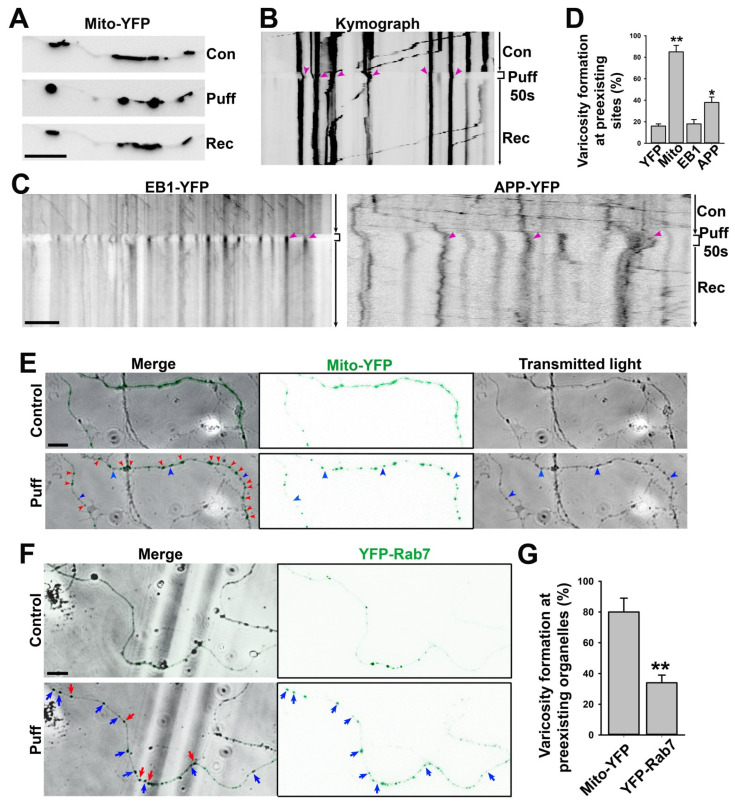
**Mitochondria are preferred sites for puffing-induced transient formation of axonal varicosities.** (**A**) An axon expressing mito-YFP before (con), during (Puff), and after puffing (Rec). YFP fluorescence signals are inverted. (**B**) Kymograph of the movement of mito-YFP puncta along the axon in (**A**). Red arrowheads, YFP puncta after puffing located at the preexisting site. (**C**) Kymographs of EB1-YFP (left) and APP-YFP (right) movement along axons before, during, and after puffing. Most YFP punta were induced by puffing at new sites and slowly resolved afterwards. (**D**) Percentage of YPF puncta after puffing located at the same sites prior to puffing. One-way ANOVA followed by Dunnett’s test: *, *p* < 0.05; **, *p* < 0.01. *n*~12 for each condition. (**E**) Live-cell imaging for mito-YFP (green) and transmitted light before and after puffing during varicosity induction. Red arrowheads, varicosities formed at preexisting mito-YFP clusters. Blue arrowheads, varicosities formed at the sites without mito-YFP. (**F**) Live-cell imaging for YFP-Rab7 and transmitted light before and after puffing during varicosity induction. Red arrows, varicosities formed in preexisting YFP-Rab7 clusters. Blue arrows, varicosities formed at the sites without preexisting YFP-Rab7 clusters. (**G**) Summary of varicosity formation at preexisting mitochondria indicated by mito-YFP clusters (*n* = 10) and MVBs indicated by YFP-Rab7 clusters (*n* = 9). Unpaired *t*-test, **, *p* < 0.01. Scale bars, 15 μm.

**Figure 5 cells-11-02533-f005:**
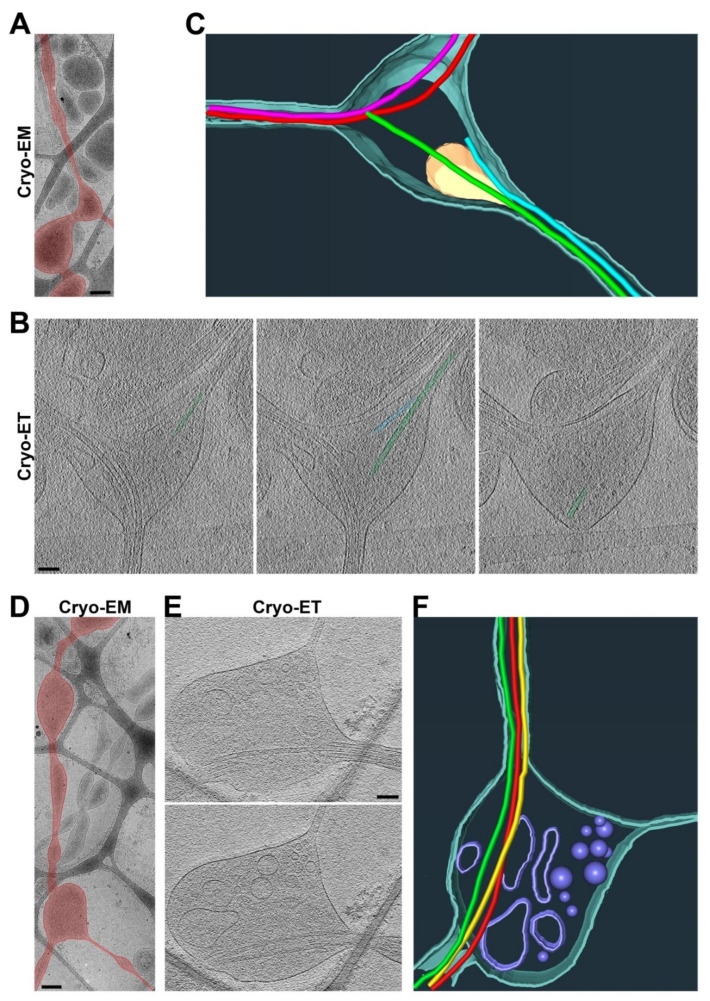
**Three-dimensional ultrastructures of two typical types of axon branch points.** (**A**) Low-magnification Cryo-EM image of an axon and its branch point. An axon with a branch point is highlighted in red. (**B**) Three slices from the Cryo-ET tomogram of the axon branch point in (**A**). De novo formation of two MTs and their free ends for one branch are highlighted in green and cyan. (**C**) Segmentation view of the branch point in (**B**). Two MTs in red extending into one of the two daughter branches. Two MTs formed de novo within the branch point (one in green and the other in light blue) and extended into the other daughter branch. Mitochondria in yellow and plasma membrane in dark cyan. (**D**) Low-magnification Cryo-EM image of an axon branch point. The axon was highlighted in red. (**E**) Two slices from the Cryo-ET tomogram of the axon branch point in (**D**). One of the two daughter branches, the thin one, did not contain a single MT. (**F**) Segmentation view of the branch point in (**E**). Three MTs (in red, yellow, and green) all extended into one of the two daughter branches. Vesicles of different sizes in yellow and plasma membrane in cyan. Scale bars, 250 nm in (**A**,**D**) and 100 nm in (**B**,**E**).

**Figure 6 cells-11-02533-f006:**
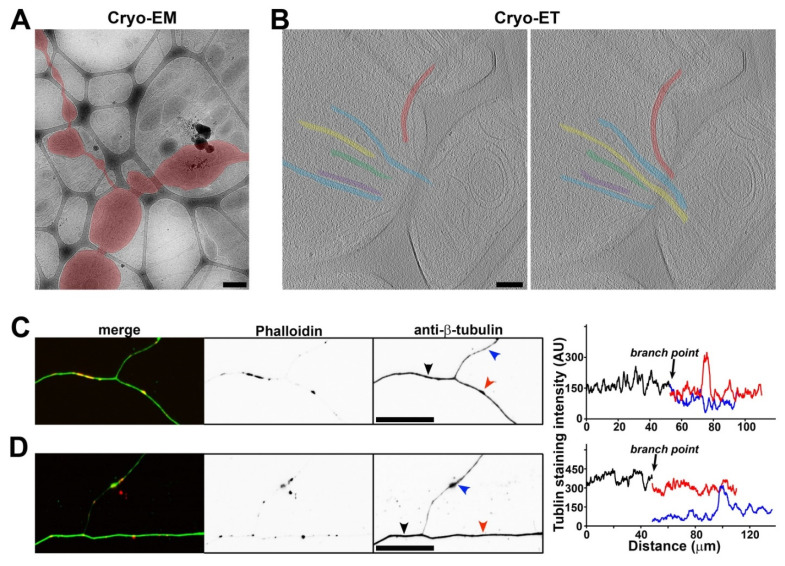
**Unequal splitting of MTs into two daughter branches.** (**A**) Low-magnification Cryo-EM image of an axon branch point. The axon is highlighted in red. (**B**) Two slices from the Cryo-ET tomogram of the axon branch point in (**A**). Five out of six MTs (in different colors) extended into one daughter branch, while one MT (in red) extended into the other daughter branch (the thinner one). (**C**,**D**) Axons growing on coated cover glass were stained with an anti-β-tubulin antibody (green) and phalloidin (red). The line profiles of β-tubulin staining intensity along axon branches were plotted on the right. Black, the main branch. Red, one daughter branch. Blue, the relatively thinner daughter branch. Scale bars, 500 nm in (**A**), 100 nm in (**B**), and 50 μm in (**C**,**D**).

**Figure 7 cells-11-02533-f007:**
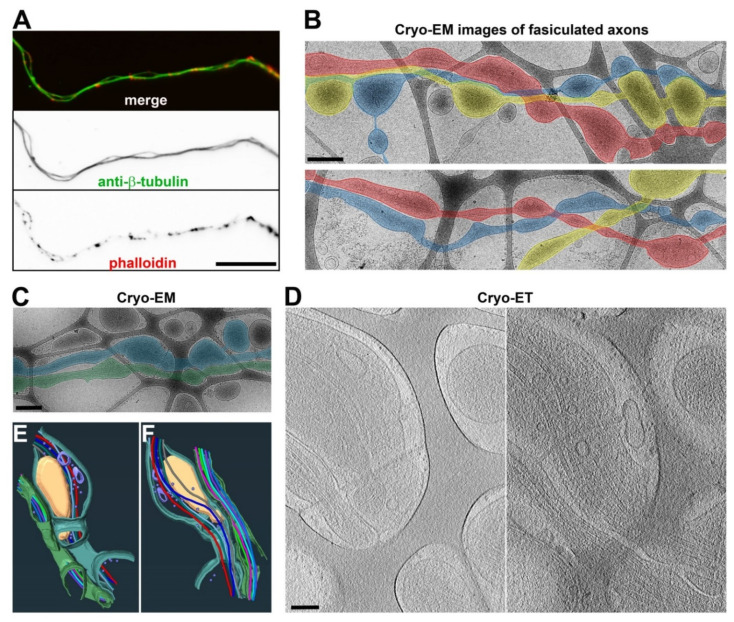
**Three-dimensional ultrastructures of fasciculated axons.** (**A**) Two fasciculated axons growing on coated glass coverslips stained with an anti-β-tubulin antibody (green) and phalloidin (red). (**B**) Low-magnification Cryo-EM images of fasciculated axons growing on top of lacey carbon film. Fasciculated axons are highlighted in blue, red, and yellow. (**C**) Low-magnification Cryo-EM image of two fasciculated axons (highlighted in blue and green) growing on top of lacey carbon film. (**D**) Two slices from the Cryo-ET tomogram of two fasciculated axons in (**C**). (**E**,**F**) Segmentation (bottom and top) views of two fasciculated axons in (**D**). MTs are shown with tubes of different colors: mitochondria in yellow, vesicles in purple, and plasma membranes in green and cyan. Scale bars, 25 μm in (**A**), 400 nm in (**B**,**C**), and 100 nm in (**D**).

## Data Availability

Not applicable.

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
