# Peer review of "The Mechanical Microenvironment Regulates Axon Diameters Visualized by Cryo-Electron Tomography"

_cells, 2022, doi:10.3390/cells11162533_

Round 1

Reviewer 1 Report

The manuscript has improved enormously with a far better narrative. I recommend publication, but would suggest that authors take the opportunity to make a few further improvements:

Abstract: The key point that should come across is that axon swellings are more likely to occur in areas of less stability, i.e. in the absence of myelination in vivo and, when cultured, in areas of less mechanical support (holes on film), or in areas where axons are pre-dilated (mitochondria) or at branch points.

l.25: What is the "novel" regulation?

l.253: might the observed varicosities on film be referred to as beading, as is used in the literature for similar swellings?

l. 329: define "asymmetric distribution of MTs"

l.331ff: not well explained. Do you mean that 4MTs stay in the primary neurite whereas MTs of the branch form de novo?

l.335: how many of your branch points contained mitochondria? Please, provide that info.

l.346: axonal branches originate from filopodia, and MTs stabilse them into a branch. See for example: https://doi.org/10.7554/eLife.50319

l.374: It would be helpful to have a brief Conclusions section.

Reviewer 2 Report

Dear Authors, 

You have answered all my questions and I do not have more comments.

Sincerely, 

Reviewer 3 Report

In this manuscript, Ma et al. identified the enlarged structures named axonal varicosities under normal conditions by combining confocal microscopy and cryo-electron tomography with in vivo and in vitro systems. Interestingly, they reported that the non-uniform mechanical interactions between cultured neurons and microenvironment can lead to 10-fold difference in axonal diameter, indicating a novel mechanical mechanism underlying the regulation in the three-dimensional ultrastructure of axons under physiological conditions.

Overall, this is a well-written manuscript containing interesting results conveyed by well-arranged figures.

Major comments:

1. When cultured on lacey carbon film coated EM grids, axons formed varicosities exclusively in the holes. This could be caused by axonal contraction on the top of the carbon fiber, but not necessarily axonal enlargement in the hole of carbon film, as the stiffness or other properties of carbon fiber may not favor axonal growth. This is consistent with the observations that axons growing on the EM grids are much thinner than those growing on the plastic surface (Fig. 1H) and in cerebral cortex (Fig. 1B and 2F). The axon shafts contract on the top of carbon fiber and push or restrict most subcellular organelles in the hole region, subsequently causing the relative enlargement of axonal diameter. While the data suggest that non-uniform mechanical interaction with microenvironment contributes to shaping three-dimensional ultrastructure of axons, this model is too artificial to be related to axonal varicosities in the brain. For example, the narrow portions of axons pose a tremendous challenge for axonal transport of large organelles, which may not be compatible with physiological conditions. The authors should discuss about this.

2. Unlike neurons growing on EM grids, the puffing pressure-induced axon varicosities are accompanied with microtubule disruption. Does it indicate a distinct mechanism involving cytoskeleton reorganization, different from the heterogeneous mechanical interaction on EM grids, but more analogous to the brain injury or disease conditions? Given that the puffing pressure-induced axon varicosities are easily formed at the sites with preexisting mitochondria, it would be interesting to monitor the dynamic change of axonal diameter at the region with or without preexisting mitochondria during live imaging (is it increased at the mitochondria sites or decreased at the other sites?), which will provide more insights.

3. The axonal cytoskeletons, including the microtubules and the periodic actin ring, play fundamental roles in regulating axon shape and branching. Along the axons growing on EM grids, the microtubules are constant in number without any disruption between axonal varicosities and nearby shafts. However, the actin organization in these regions are still unknow. Within the axonal varicosities, the actin organization is presumably altered or more dynamic, which may contribute to axonal branching via easily forming actin-based protrusions for microtubules to invade. Do the authors have any direct data on this? If not, it would be important to discuss about this.

4. The immunostaining of β-tubulin and actin in Fig. 6C, 6D and 7A needs to be further analyzed, especially for actin and β-tubulin hotspots in axon shaft and branch. The authors may want to examine neurons with axonal varicosities as well (induced by non-uniform mechanical interaction or fluid puffing) to check the correlation between the hotspots and axonal varicosity formation.

Minor comments:

1. The example in Fig. 1A appears not be consistent with the statistics. Axons in the corpus callosum exhibit more significant diameter variations than axons in the cortex (Fig. 1B).

2. In Fig. 1B, the ratio of Dmax/Dmin = 13.1 may represent an extreme value (perhaps corresponding to the bottom right dot) and most of the other ratios are much smaller. The authors should check this.

3. The statements in line 179 “many of them are myelinated” and line 191-192 “most of these varicosities presumably represented presynaptic boutons” are not supported by experimental data or citations.

4. The Discussion section is too long, containing many repetitive statements from the Introduction and Results section.

Author Response

This manuscript is a resubmission of an earlier submission. The following is a list of the peer review reports and author responses from that submission.

Round 1

Reviewer 1 Report

In this study, the authors provide beautiful and carefully implemented analyses of axons in vivo and in culture, but they fail to bring across a convincing point they want to make, and that would explain why their study is important. The in vivo work is linked to the culture work through a simple correlation that is far-fetched (see my comments below), and the branching point section does not fit into the overall story at all. I do not see this work being publishable in its current form, but might see potential if the story line is significantly changed. For example, you could pose the question as to whether diverticula formation is facilitated in conditions of low membrane rigidity which you test in vivo (myelinated versus non-myelinated - although the complication of potential synaptic boutons makes interpretation difficult) and in culture (bundled versus single axons). How your transport and branching stories fit into this, I would not know at this point.

Detailed comments:

l.13: a rather vague statement. Please, specify what property you refer to that is not understood, hence what you intend to do to clarify this point.

l.16: specify "non-uniform"

l.18. explain the rationale for using Thy1-YFP

l.20: What are these holes? Explain

l.22: explain 'constant'

l.23: sentence entirely unclear; 'asymmetrically separated'? 'de novo formation': does this refer to nucleation?

l.25: what are the novel features?

l.32: add RNA

l.40-42: specify "diameter variations" to substantiate the significance statement

l.44: do "axonal varicosities" refer to "diameter variations"? I get the impression you may talk here about swellings or diverticula?

l.52: explain the link from MT depolymerisation to varicosity formation

l.66: Explain what you mean with "structure"

l.72f.: statement does not align well with previous sentences. Is the whole section about actin rings relevant here in the intro or should it better be mentioned in the discussion?

l.165: please, define "High-resolution" with respect to the resolution that can be achieved in your confocal system. For example, can you provide voxel sizes?

l.180ff. As the authors state, there are potential synaptic boutons in the grey matter and of course the stiff myelin sheath in the CC needs to be considered, so the discussion of impact of the environment appear almost trivial and might better be taken out.

Fig.1G/H - there is a numbering error

l.233: I very much disagree that the (potentially coincidental) correlation in varicosity size establishes a mechanistic link between the in vivo and culture situation. Low membrane rigidity in cultured, non-fasciculating and non-ensheathed axons will be prone to respond to areas of low mechanical support. This is a situation very different from the brain where cell-cell interaction is involved.

l.243: did you ever observe axons without MT in vivo? Are you sure you looked at true axons? Taking away the transport highway is a pathological condition.

l.274: The EB1 experiment requires more explanation. EB1 is visible only in comets upon MT polymerisation which is, by default, combined with tip extension, i.e. movement. What would those stationary Eb1 pools be? This is an odd finding that should either be removed or properly explained.

l.280 what is YFP, as compared to YFP+? I don't understand

l.281: if transported APP cargo is trapped at a newly formed varicosity, why is this the case? Do you observe MT disturbances upon puffing (unlike at holes without puffing)?

Section 3.3.: whilst I can follow the mitochondria and MVB story, I am lost with the EB1 and APP stories.

Fig.5B: the demonstration of how MT branch is beautiful.

l.311: the non-attached MTs are not shown but should be. In the top image of E one might suspect an MT, but this is not enough. Otherwise, it could be considered a filopodium.

l.315: please, explain what you would consider "MT branching" (MT bundle branching?); Fig.5B appears a perfect case likely reflecting a nucleation event on the lattice of the existing bundle. This is the common view of how branching will occur, or it might involve severing and reorientation (the second type of branching you observe?)

l.320: MTs in axons are not too rigid and can curl with a diameter well below 1 micron. They are softened by kinesins and through acetylation. See also: https://doi.org/10.1083/jcb.201912081 and https://doi.org/10.1186/s13064-019-0134-0

Fig.6: this might be the third form of branching where existing MTs of the bundle split up into different branches. However, this is not well shown in the image.

l.332: more intense actin might be trails or waves, both of which have been described in the literature; or they may be swellings, but the resolution is not good enough.

Section 3.5 is expected at argued above, especially the fact that varicosities occur on the outside.

The discussion does not add much to the paper and does not clarify as what the the authors consider the main outcome of the work and why it is important.

Reviewer 2 Report

Dear Authors

Overall, it is an interesting work with nice results, and It only has some minor points.

1.- Describe the age and sex of adults mice and provide the number of IACUC for this procedure. 

2.- In adult mice which part of the cortex did you analyze?

3.- There is an arroba with tubulin instead a beta. And in line 124:  with ~97 m grid hole size. In line 205: axons with diameters ≤ 2 m. 

4.- The number of samples sometimes is in capital letters (Figure 2) and sometimes in lower case (Figure 4) and in the others is missing. In Figure 4: n ~ 12 meand aprox 12?

King regards

Round 2

Reviewer 1 Report

I am rather disappointed by the revisions of the manuscript, and that authors responded with minimal effort only by responding to detailed comments.

I am deeply interested in this area of axon biology. However, like in the first version, I do not feel that I take any message away from the revision - in spite of the beautiful experimental data provided. Not even the title or abstract convey a clear message that would attract me as a reader. The key problem posed in the abstract is: "How these long slender structures are mechanically regulated remains poorly understood" - this does not pose a precise cell or neurobiological problem and lacks a clearly presented relevance. Unfortunately, this is not compensated for at the end of the Introduction or the start of the Discussion, and the work therefore seems mainly driven by experimental rather than scientific aims. The analyses of branch points still enter this work in a rather casual fashion which does not fit the concept of microenvironmental impacts, interpretations of roles of MTs are questionable, whereas important discussion about membrane rigidity are absent.  

We live in times where journals get flooded with publications, and it is more important than ever that authors invest deep thought about the relevance of their work and apply hypothesis-driven conceptual approaches to avoid being overlooked. In my view, the authors have not taken this opportunity to embed their work in a relevant enough context that would grant publication.